

# Variation in calcification of *Reticulofenestra* coccoliths over the Oligocene-Early Miocene

José Guitián[1,2*◊], Miguel Ángel Fuertes[3*], José-Abel Flores[4], Iván Hernández-Almeida[1], Heather Stoll[1]

[1] Dept. of Earth Science, ETH Zürich, 8092 Zurich, Switzerland

[2] *Now at*: Centro de Investigación Mariña, Universidade de Vigo, GEOMA,  Vigo, 36310, Spain

[3] Dept. of Didactics of Mathematics and Experimental Sciences, University of Salamanca, Salamanca 37008 Spain

[4] Dept. of Geology, University of Salamanca, Salamanca, 37008, Spain

[◊]*Correspondance to*: José Guitián[1] (jose.guitian@uvigo.es) and Heather M. Stoll (heather.stoll@erdw.ethz.ch)

*These authors contributed equally

**Abstract**

Coccolithophores are calcifying marine phytoplankton whose intracellularly produced calcite plates, or coccoliths, have been the dominant source of calcium carbonate in open ocean settings since the Cretaceous. An open question is whether their calcification has been affected by variation in environmental parameters such as the ocean carbon system over geological

timescales. Previous methods using circular polarized light microscopy allowed only the thickness of small coccoliths thinner than 1.5 microns to be quantified but prior to the Pliocene, a significant fraction of the coccoliths exceed this thickness and have not been quantifiable. Here, we implement a new approach for calibration of circular polarized light microscopy enabling us to quantify coccoliths which feature calcite up to 3 microns thick. We apply this technique to evaluate the evolution of calcification in the *Reticulofenestra* from the Early Oligocene to Early Miocene in exceptionally

well-preserved sediments from the Newfoundland margin. Through this time interval, coccolith thickness and the scale-invariant shape factor $k_{se}$ vary by about 20% around the mean thickness of 0.37 μm and mean $k_{se}$ of 0.16. Lower shape factors characterize samples with higher relative abundance of dissolution-resistant nannoliths, suggesting that dissolution may contribute to thinning of placoliths. We therefore define temporal trends in calcification only in samples in which the assemblage suggests minimal dissolution. Lowest $k_{se}$ characterizes the Middle Oligocene, and highest $k_{se}$ around 18 Ma in the

Early Miocene. High ocean DIC concentrations have been proposed for this period of the Miocene, and may be one factor contributing to high coccolith $k_{se.}$

## 1. Introduction

Coccolithophore algae possess small calcareous plates called coccoliths, produced continuously intracellularly to maintain a full interlocking covering over the cell. Calcification by coccolithophorid algae represents the major source of $CaCO_3$

production in the open oceans (Siesser, 1993). The abundance, growth rate, and degree of calcification of coccolithophores in the ocean have implications for the global carbon cycle. Calcification is important because it affects the surface ocean alkalinity, and may affect the sinking velocity of coccolithophores  and ballasting efficiency of organic carbon export (Rost and Riebesell, 2004).

Calcification rates of coccolithophores depend on both the cellular growth rate and the degree of cellular calcification, which

to first order is controlled by the thickness of coccoliths (Bolton et al., 2016). In laboratory culture studies, the cellular calcification is generally measured by geochemical assay of calcite (either as Ca or inorganic C) in a culture volume, paired with estimation of the cell density. Multiple laboratory studies of extant coccolithophores strains show plasticity in calcification rates and cellular calcification in response to environmental  conditions, especially changing $CO_2$ and media



carbon chemistry, as reviewed (Bach et al., 2015). It remains uncertain if this plasticity in short term culture is
representative of the longer term response of calcification to variations in atmospheric $CO_2$ and ocean carbon chemistry in
the geological past. However, variation in cellular calcification by past coccolithophore communities in the ocean can
usually be assessed only from the individual disaggregated coccoliths present in deep-sea sediments. Hence, there is
significant interest in quantifying variations in their cellular calcification by quantifying variation in the amount of calcite in
coccoliths over time in the past.

In this contribution, we seek to evaluate changes in cellular calcification in coccoliths from the Mid-Oligocene through Early
Miocene in a deep-sea sediment series of exceptional preservation quality from coring in IODP Site 1406 on the
Newfoundland Margin. This time period has been inferred to feature a significant decline in $pCO_2$ (Zhang et al., 2013).
Significant changes in the mean size of coccolithophores during the Oligocene have also been proposed as an adaptation to
declining $CO_2$ (Henderiks and Pagani, 2008; Guitián et al., 2020). The Noelaerhabdaceae lineage, which includes the recent
genera *Emiliania*, *Gephyrocapsa*, *Pseudoemiliania, Reticulofenestra*, have dominated the coccolithophore communities over
the Cenozoic. For this reason, we focus on the record of changing cellular calcification within the Noelaerhabdaceae family,
commonly known as "reticulofenestrids" through the Oligocene-Miocene (Young, 1998). Because Oligocene to Miocene
sediments include many large coccoliths from the *Reticulofenestra* genus (Guitián et al., 2020) they feature portions of
coccoliths thicker than the limit of published techniques for estimating coccolith thickness by polarized light microscopy
(1.55 μm (Bollmann, 2014; Fuertes et al., 2014); 1.7 μm (Beaufort et al., 2021); and 1.34 μm (Johnsen and Bollmann, 2020).
Consequently, we implement here a new system to quantify populations of thicker coccoliths by automating the calibration
approach of (González Lemos et al., 2018).

**2. Approaches for estimation of coccolith thickness**
Estimations of coccolith mass were first derived from volumes of rotation using idealized three dimensional cross sections of
type specimens produced from scanning electron microscope images (Young and Ziveri, 2000). This approach generated a
single "shape factor" ($k_s$, see Equation 1) for a given species or morphotype, and the volume was calculated as the product of
the shape factor and measured length. However, the technique was not suited for quantifying variation in thicknesses of
individual coccoliths of a given length. Polarized light microscopy was subsequently proposed to directly estimate the
thickness of individual coccoliths because thicker coccoliths generated greater birefringence. The first technique used linear-
polarized light with images captured over multiple rotations of the stage to estimate the full calcite volume (Beaufort, 2005).
Later, circular polarized light was used to eliminate extinction patterns by making the birefringence independent of the angle
of the stage, so estimations of coccolith volume could be made from a single image and therefore with greater precision
(Fuertes et al., 2014; Bollmann, 2014). Such methods have shown that cellular calcification of small coccoliths has been
decreasing since the Late Miocene (Bolton et al., 2016).

In all polarized light methods, the calibration of the colour or grey level of each individual pixel in the image to absolute
thickness is a challenge. To date, calibration challenges have limited quantification of coccolith thickness to the smallest
coccoliths with thin calcite units; typical thickness limits are : 1.7 μm (Beaufort et al., 2021) and 1.34 μm (Johnsen and
Bollmann, 2020). A recent calibration approach proposed the use of a calcite wedge in which the thickness is independently
determined at various points using a tilting compensator that provides known retardation of light rays (González Lemos et
al., 2018). The main advantages of the calibration with calcite wedge is that it is independent of the type of microscope and
its light configuration, and color-thickness calibration is in theory possible up to 4.5 μm. As coccoliths with a thickness
greater than 1.55 μm appear with colours ranging from orange to blue, this calibration can be applied to all the coccoliths
beyond the previous grey-level limit.



For this study, the Oligocene to Miocene sediments include many large coccoliths from the *Reticulofenestra* group,
following (Young, 1998), which feature portions of coccoliths thicker than 1.55 μm, we therefore devise a system for
automating the calibration approach of (González Lemos et al., 2018) within the range of thickness from 0 to 3 μm. We use a
new calibrated calcite wedge and code the C-Calcita program (Fuertes et al., 2014) to employ this calibration and
subsequently to calculate thickness in each pixel of coccoliths – reaching 3 μm in many studied coccoliths – and thereby
compute the mass and average thickness of the individual coccoliths. This calibration can be directly applied to estimate
absolute thickness of coccoliths within the Noelaerhabdaceae lineage. Noelaerhabdaceae are formed of calcite units with a
radially oriented c-axis (r-units) without appreciable elements with vertically oriented c-axis (v-units) (Young et al., 1992).
R-type units are fully visible when circular polarized light is used.

Since v-units are not fully visible in circular polarized light, relative, but not absolute, calibration of thickness is possible for
taxa which contain significant r- and v-calcite units (e. g. (Fuertes et al., 2014). Nonetheless, relative variations in thickness
of other taxa may be still relevant to studies of coccolith preservation and diagenetic overgrowth (Dedert et al., 2014).
Consequently, we also report the estimation of differences in thickness of other taxa.

### 3. Methodology for estimation of coccolith thickness

#### 3.1 Microscope setup for image acquisition

All the images were obtained at ETH Zurich using a Zeiss Axio Scope HAL100 POL microscope with circularly polarized
light, equipped with a Zeiss Plan-APOCHROMAT 100x/1.4 Oil objective, a universal condenser with numerical aperture set
at 0.1-0.2. Circular polarization is obtained placing two λ/4 retardation plates, one between the lower linear polarizer and the
condenser and the other below the upper linear polarizer, both at an angle of 45° relative to the linear polarizers. The system
is equipped with a Zeiss Axiocam 506 Color camera. Light intensity and camera exposure is defined checking calcite and
coccoliths fields of view briefly, before each calibration session starts to avoid overexposure of RGB colours, and gamma
correction factor is set to 1. Camera resolution was 0.045 μm per pixel, with final images of 2560x1920 pixels.

#### 3.2 Calibration of absolute thickness within C-Calcita

A calcite wedge with a constant slope over this thickness range has been produced at ETH Zurich. The calcite thickness was
measured in 11 points along a linear profile, using a tilting compensator, as described in (González Lemos et al., 2018)
Figure 1. We refer to this "reference image" as Cal.1, and the coordinates of the points whose thickness has been measured,
Coord1.

To account for differences in microscope configuration or light situation, during each microscope session in which coccolith
images are taken, we take a new "session image" of the calcite wedge, seeking to capture as similar a field view as possible
to the reference wedge image, and using microscope settings identical to those used for capture of coccolith images.

With Matlab®, the original C-Calcita script (Fuertes et al., 2014) was modified to an updated version, C-Calcita v.2, where
we establish a routine to locate the calibrated thickness points from the reference wedge image into each session photo (e.g.
Cal2) (Figure 1). The relationship between both the reference and session images of the wedge is searched and located the
equivalent points. This routine employs the KAZE detection and description, which was designed to find point
correspondences between two images of the same scene (Alcantarilla et al., 2012). This routine occurs in three steps: first,
'interest points' are selected at distinctive locations in the image, such as corners, blobs, and T-junctions. Then, a feature
vector, called descriptor, represents the neighbourhood of every interest point to finally march descriptor vectors between the
different images, in order to obtain the rotated angle and the translation done. Subsequently, the position of the known





coordinates of the reference image Cal1 can be located on the session image Cal.2 (Figure 1C). Because the wedge can feature cleavage breaks at the edge, we calculate the theoretical point in the image that would correspond with zero thickness using a polynomial fit from the measured points, and then calculating the roots for that polynomial.

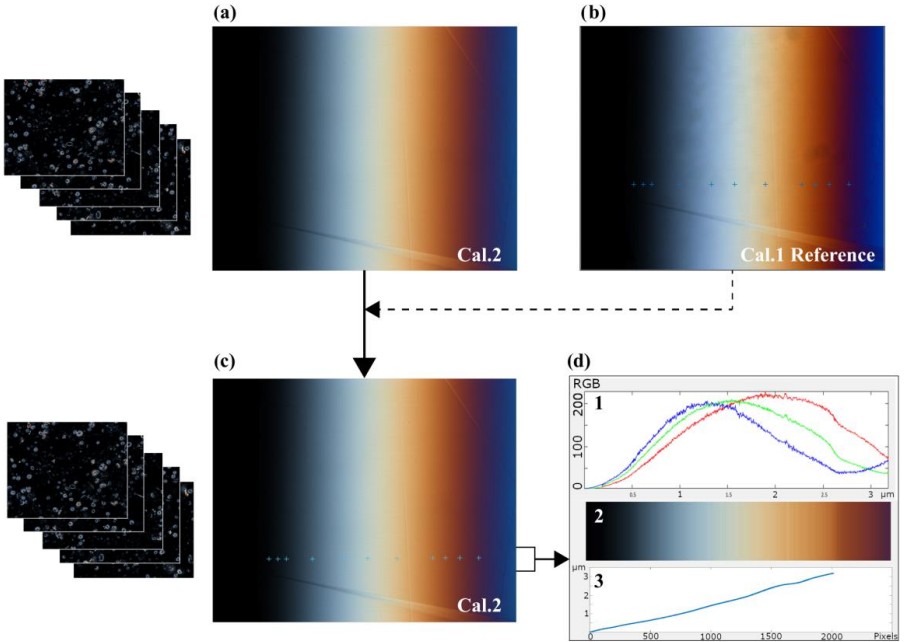


**Figure 1.** (**a**) Session image of reference calcite wedge (Cal.2) taken with the same microscope settings than the coccoliths images that are going to be taken later. (**b**) Reference image of the calcite wedge (Cal.1) and the points where the thickness has been measured using a tilting compensator (Coord1). Notice although similar, nor frame nor light it exactly the same as the image Cal.2. (**c**) Session image Cal.2 with the automate calculated equivalent points of known thickness. (**d**) 1. Color band spectrum along the line profile. 2. RGB color rectangle showing in lines each point line color. 3. Thickness profile along the wedge line.


Once the known thickness calibration points have been identified on the session image, the C-Calcita routine generates a matrix containing the RGB values of all the pixels of the image Cal.2 along the profile given by the points whose thickness has been measured, up to the calculated point that would correspond to zero thickness. As illustrated in Figure 1, this process has interpolated between the 11 known points to generate a calibration set in which more than two thousand points between 0 to 3.2 microns have a defined thickness value and set of RGB values for the microscope and setting employed (Figure 1D first panel). We describe this matrix as a RGB-Thickness calibration matrix, which provides the complete color calibration of C-Calcita for a particular microscope session (Figure 1D third panel).


### 3.3 Calculation of thickness of coccoliths


Slides of coccoliths are prepared using the decanting method described by (Flores and Sierro, 1997) which allows for random settling of coccoliths. Mounted slides are then inspected on the microscope and sample images are taken using settings described in 3.1.





Once the calibration is completed, the coccolith images that have been captured with the same configuration as the image Cal.2 for reference are loaded into the program. In order to measure the coccolith thickness that corresponds with to each pixel in the coccolith images, an exhaustive search is created based on the euclidean distance: for each individual pixel of the coccolith image, the most similar RGB vector of the calibrated matrix is matched. The thickness value assigned to that single coccolith pixel is that of the most similar RGB matrix vector.

In some cases, the internal reflections of the light into the coccolith, or the presence of aberrations may lead to inference of a higher than realistic thickness in some points (e.g. Figure 2). To minimize this issue, a gaussian filter is applied to the coccolith image, obtaining a smoothed thickness image. In those pixels where the obtained thickness increases abruptly, compared with the smoothed thickness image, the thickness value for that pixel is substituted by the gaussian value of its surroundings (Figure 2).

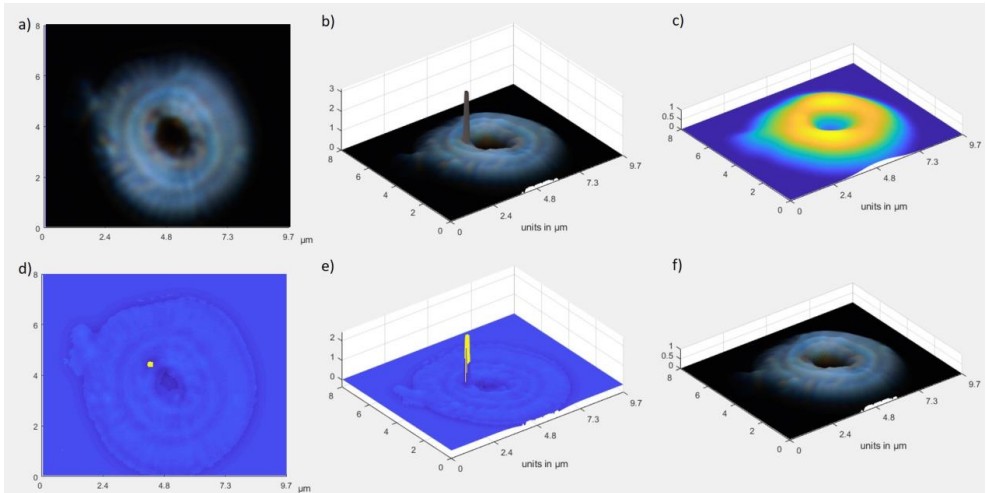

**Figure 2.** (**a**) Original image; (**b**) 3D Coccolith with thickness in some points higher than expected; (**c**) Gaussian smooth; (**d, e**) identification of points to be smoothed (yellow); (**f**) Final 3D image where the identified points have been smoothed.

Finally, from the smoothed map of thickness per pixel, the total area-integrated thickness is used to compute the volume and mass of calcite in each coccolith, as in previous versions of C-Calcita (Fuertes et al., 2014).

**3.4 Indices of variations in thickness and calcification of Oligocene-Early Miocene coccoliths**

Eleven samples from IODP Site 1406, which cover the interval between Oligocene to Early Miocene (27.9 to 17.8 Ma), were analyzed for coccolithophore thickness. The age model applied for that interval was the one previously published by (Guitián et al., 2019)). Between 241 and 660 coccoliths per sample were isolated from microscope images and major and minor axis were calculated, and the total calcite mass computed. In one additional sample, only 86 coccolith images were obtained and

processed. Total coccoliths/nannoliths in all sites included 55 *Discoaster*, 83 *Helicosphaera*, 96 *Sphenolithus*, 154 *Coccolithus*, and 3018 within the *Reticulofenestra* genera (Young, 1998), including the *Cyclicargolithus* group.

As described previously (Fuertes et al., 2014), the C-Calcita estimates for each identified coccolith, the major and minor axis, the cocccolith area, and the coccolith volume. From the major and minor axes, the program estimates circularity, as



well as thickness (calcified volume/calcified area), and converts coccolith volume into coccolith mass using the density of calcite. Hereafter, we refer to the major axis as length (l) and the minor axis as width (w).

We calculate several additional parameters including an ellipsoid defined by the coccolith length and width, the thickness (volume/ area), and two shape factor indices. For a given coccolith geometry, a change in the coccolith major and minor axis will result in a thickness which covaries with mass. Therefore, thickness could be expected to correlate with coccolith length. In contrast, the shape factor indices are independent of variations in coccolith size.

The shape factor $k_s$ has been defined (Young and Ziveri, 2000) as an estimator of the fraction of a cube of length (l), which is filled with the volume (v) of the coccolith:

$$k_s = \frac{v}{l^3} \qquad (1)$$

where for a sphere $k_s$ is 0.52. Many coccoliths are often elliptical, but the major axis length has been used in the standard definition of $k_s$ (Young and Ziveri, 2000).

We also calculate a shape factor referenced not to a cubic volume but rather the volume of an ellipsoid defined by the coccolith l and w, ($k_{se}$) which allow calculation of a shape factor independent of variations in circularity:

$$k_{se} = \frac{v}{\frac{4}{3}\pi \frac{l}{2}(\frac{w}{2})^2} \qquad (2)$$

## 4. Results and Discussion

### 4.1 Thickness of Oligocene nannofossils

In the thickest part of the coccolith of the largest *Reticulofenestra*, thicknesses reached 3 microns, spanning the full range of the implemented color calibration (Figure 3). As observed in Figure 3, the thickness maps suggest minor imperfections in preservation and breakage. We examine these in the statistics of data from large populations, which minimizes the influence of these breakages on the estimation of coccolith thickness from sediments of a given age.

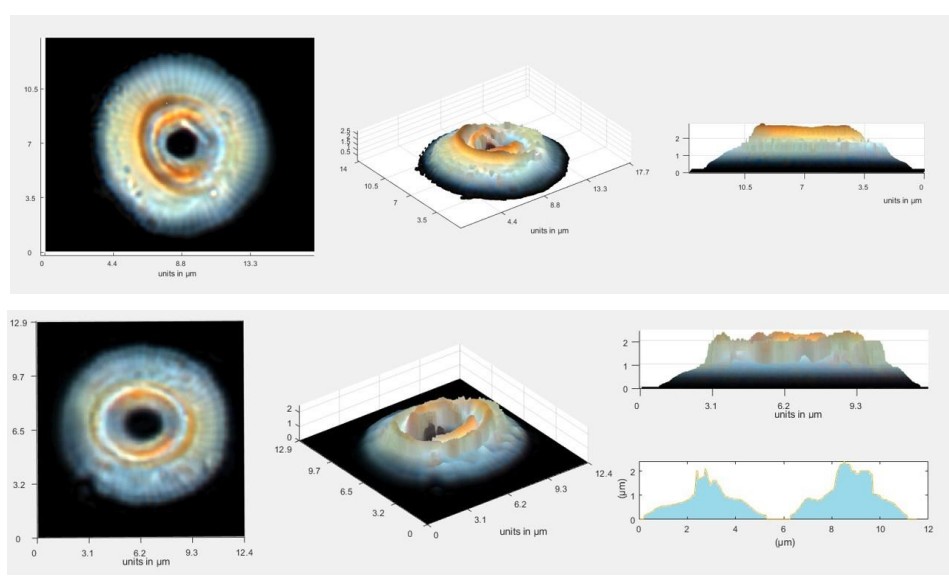

**Figure 3.** Example reconstructions of large *Reticulofenestra*.



Averaged over the entire calcified area of the coccolith, 95% of *Reticulofenestra* coccoliths had thicknesses in the range of 0.26 to 0.72 microns (Figure 4). Among these "reticulofenestrids", 95% of the coccoliths had major axis between 3 and 8.6
190    microns, as expected from a previous study (Guitián et al., 2020). Modern *Reticulofenestra* are generally smaller than those we measured from Oligocene-Early Miocene. A survey of 3600 *Gephyrocapsa spp.* in core top sediments range from 0.9 to 6.2 µm, with the mean lengths at a given site ranging from 1.3 to 4.2 µm, with lengths over 3.5 µm restricted to warmer waters >22°C (Bollmann, 1997).

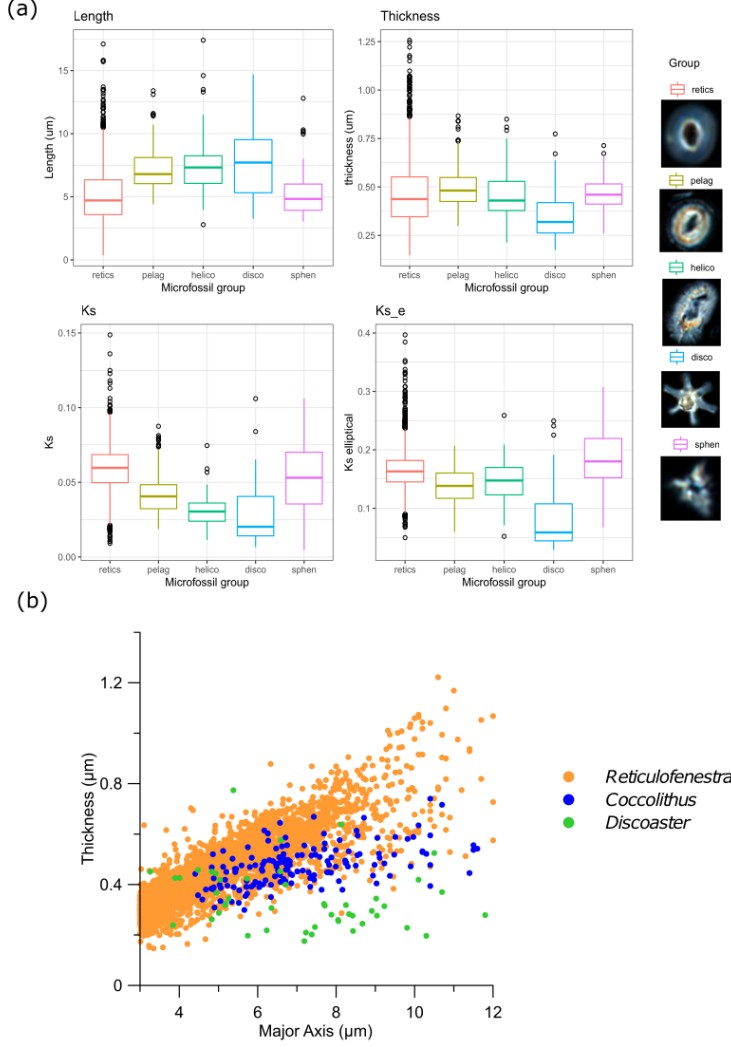

195    **Figure 4.** (**a**) Box and whisker plot showing length, thickness, and ks for taxa in all samples and thumbnail examples of *Discoaster*, *Sphenolithus*, *Coccolithus* and *Reticulofenestra*; (**b**) plot of thickness vs major axis for all *Reticulofenestra*, *Coccolithus*, and *Discoasters*.





The calculated $k_s$ and $k_{se}$ values are lower for the placoliths with mixed r- and v- units (*Helicosphaera* and *C. pelagicus*; 0.03
and 0.04, respectively) compared to the *Reticulofenestra* group with r-units comprising the full coccolith (0.06) (Figure 4c).
Similarly, the slope of thickness vs length curve is also lower for *Helicosphaera* spp. and *C. pelagicus* compared to
*Reticulofenestra* (Figure 4c). We suggest that these results are an expected consequence of the crystallographic orientation.
Previous estimations of $k_s$ made not from polarized light but from volumes of rotation of SEM cross sections, which are
independent of crystallographic orientation, indicate $k_s$ values for recent *C. pelagicus* and *Helicosphaera* spp. (0.06, 0.05,
respectively) comparable to or larger than those of modern *Reticulofenestra* (0.02 to 0.05) (Young and Ziveri, 2000). We
conclude that the higher $k_s$ in Oligocene to Early Miocene *Reticulofenestra* than in other placoliths reflects the expected
underestimation of absolute thickness by polarized light for coccolith genera featuring significant v units. Nannoliths
(*Discoaster, Sphenolithus*) measured here have no modern or recent counterpart with which to compare the estimated $k_s$ but
likely also have underestimation of thickness. For all these non- *Reticulofenestra* taxa, we suggest that time series in future
studies be interpreted merely as relative changes in thickness and $k_s$. Alternatively, the thickness of the radial central tube
may be quantified (Cubillos et al., 2012). In this study, due to the low average number of counted nannoliths and non-
*Reticulofenestra* placoliths per sample, we do not interpret time series trends.

**4.2 Relationships between calcification and cell size within measured *Reticulofenestra* populations**

Within each of the 11 Oligocene to Early Miocene populations of *Reticulofenestra* of a given age, there is a large range in
length from ~ 3 to 10 μm. In each sample, across this range in length, the $k_{se}$ remains stable with size, or features a slight
increase in $k_{se}$ with decreasing length (Figure 5). From cultures of modern *Reticulofenestra*, the generally larger modern *G.
oceanica* coccoliths (4 μm) have higher $k_s$ than the on average smaller (3 μm) *E. huxleyi* (Bolton et al., 2016). But this
relationship is not universal (Young and Ziveri, 2000) and we are aware of no $k_s$ determinations on large populations of
extant Noelaerhabdaceae from marine sediments, because mass, rather than thickness or shape, is commonly reported
(Beaufort et al., 2011). Among a wide range of extant placolith genera ranging in length from 3 to 10 μm, there is a trend of
2 to 4-fold higher $k_s$ among larger coccoliths (Young and Ziveri, 2000; Bolton et al., 2016). Yet, whatever pressure favors an
upwards adjustment of $k_s$ with increasing size across modern placolith taxa, this factor was apparently not favoring such an
adjustment of $k_s$ within the *Reticulofenestra* across a comparably large range of cell sizes in the Oligocene.

In our Oligocene samples, the coccolith thickness increases linearly with the coccolith length in all populations and the slope
of this increase is statistically indistinguishable.

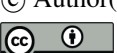



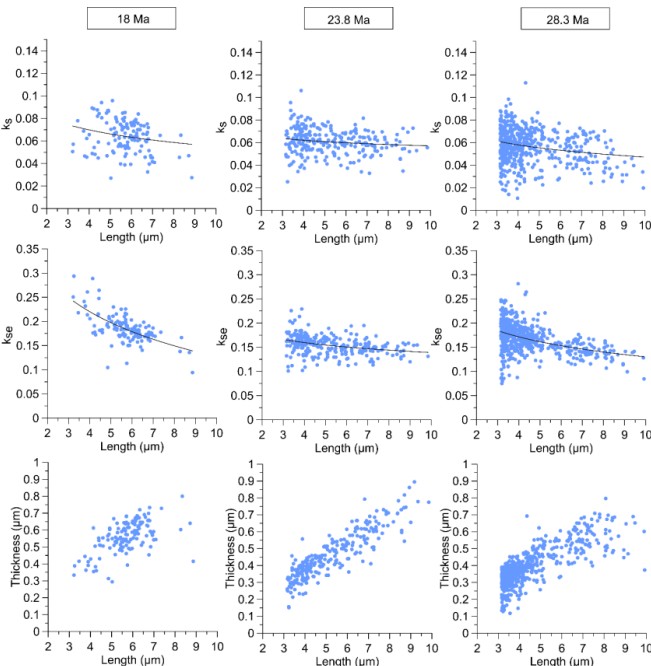

**Figure 5**. Relationship among ks, kse and thickness with length, for reticulofenestrid populations from samples of three representative ages.

### 4.3 Indicators of coccolith calcification and relationship to preservation

To evaluate temporal trends through the Oligocene to Early Miocene, we assess the median morphometric parameters of each of the 11 analyzed samples (Table 1). We examine the median as an index which is very insensitive to outliers. The median differs little from the mean, and the large (>200) populations examined yield relatively narrow confidence intervals around the mean.

The median shape factor calculated relative to a cube ($k_s$) and the shape factor calculated relative to an ellipsoid ($k_{se}$) are 235 highly, but imperfectly correlated (Table 2; r=0.76), suggesting that the $k_s$, is influenced by changes in the degree of circularity of the coccoliths over the sample set. Consequently, hereafter we rely predominantly on the $k_{se}$ metric.

Because of significant temporal trends in the size of "reticulofenestrids" (Guitián et al., 2020), temporal variations in the thickness of coccoliths may reflect changes in population mean size. The temporal variations in thickness are uncorrelated with the shape factors $k_s$ or $k_{se}$ (r <|0.05| and p>0.1), suggesting different controls on the two indices. Trends in thickness of 240 the whole size population are dominantly an expression of changing coccolith length; the temporal changes in thickness are highly correlated (r=0.85) with variation in the coccolith length. Even when considering only the population of coccoliths with lengths between 3.5 and 4.5 μm, the temporal trend in thickness still shows significant modest correlation (r=0.63) with the length. For this subset the common effect of shape on thickness is evident: the thickness of coccoliths is modestly correlated with the $k_{se}$, and strongly correlated (r=0.81) with the $k_{se}$ of the full population of coccoliths.






**Table 1.** Samples, ages, and median morphological parameters.

| Site | Hole | Core | | Section | Half | Interval Top | Interval Bot | Depth (mbsf) | Age (Ma) | ks | kse | Thickness (µm) | Length (µm) | Circularity | % (Dis+Sph)/(Ret+) | CaCO3 % | NAR (coccoliths *cm2) |
|---|---|---|---|---|---|---|---|---|---|---|---|---|---|---|---|---|---|
| 1406 | A | 2 | H | 5 | W | 36 | 40 | 12.58 | 18 | 0.066 | 0.191 | 0.538 | 5.27 | 0.902 | 1.40 % | 43 | 6139 |
| 1406 | A | 3 | H | 2 | W | 4.5 | 8 | 12.58 | 18.4 | 0.05 | 0.157 | 0.493 | 5.93 | 0.894 | 3.70 % | 44 | 4894 |
| 1406 | A | 3 | H | 5 | W | 124 | 128 | 22.838 | 18.8 | 0.048 | 0.143 | 0.509 | 6.59 | 0.901 | 4.10 % | 46 | 5223 |
| 1406 | A | 4 | H | 3 | W | 81 | 85 | 29.028 | 20.7 | 0.065 | 0.173 | 0.458 | 4.51 | 0.917 | 1.00 % | 45 | 22546 |
| 1406 | A | 6 | H | 6 | W | 74 | 80 | 52.47 | 22 | 0.061 | 0.158 | 0.505 | 5.5 | 0.929 | 2.20 % | 43 | 14357 |
| 1406 | A | 9 | H | 6 | W | 91 | 95 | 81.158 | 23.1 | 0.062 | 0.175 | 0.366 | 3.6 | 0.908 | 0.80 % | 49 | 68828 |
| 1406 | A | 10 | H | 7 | W | 37 | 41 | 91.288 | 23.8 | 0.061 | 0.155 | 0.426 | 4.66 | 0.934 | 1.60 % | 42 | 14656 |
| 1406 | A | 13 | H | 3 | W | 97 | 103 | 114.7 | 25.3 | 0.058 | 0.147 | 0.449 | 5.06 | 0.926 | 2.60 % | 40 | 26287 |
| 1406 | A | 14 | H | 4 | W | 97 | 103 | 125.7 | 25.8 | 0.058 | 0.15 | 0.453 | 5.16 | 0.928 | 1.80 % | 35 | 23151 |
| 1406 | A | 16 | H | 4 | W | 26 | 30 | 143.978 | 26.6 | 0.057 | 0.158 | 0.385 | 4.28 | 0.91 | 1.10 % | 39 | 52945 |
| 1406 | A | 16 | H | 6 | W | 66 | 70 | 147.3775 | 28.3 | 0.058 | 0.169 | 0.373 | 3.83 | 0.901 | 0.60 % | 40 | 34144 |





**Table 2**. Cross correlation of temporal changes in median geometrical indicators for the 11 studied intervals.

| | Median ks Y+Z | Median ks ellip | Thickness per ellip. area | Thickness per calc. area | Major axis | Circularity | Calc. area/ellip. area | Nannolith/retic+cocco | %CaCO3 | NAR |
|---|---|---|---|---|---|---|---|---|---|---|
| *Median ks Y+Z* | | | | | | | | | | |
| *Median ks ellip.* | 0.76 | | | | | | | | | |
| *Thickness per ellip. area* | -0.02 | 0.01 | | | | | | | | |
| *Thickness per calc. area* | -0.04 | -0.02 | 1.00 | | | | | | | |
| *Major axis* | -0.51 | -0.44 | 0.85 | 0.87 | | | | | | |
| *Circularity* | 0.41 | -0.26 | -0.02 | -0.02 | -0.12 | | | | | |
| *Calc. area/ellip. area* | 0.55 | -0.08 | 0.11 | 0.10 | -0.10 | 0.89 | | | | |
| *Nannolith/retic+ cocco* | -0.76 | -0.64 | 0.59 | 0.61 | 0.89 | -0.22 | -0.27 | | | |
| *%CaCO3* | 0.05 | 0.34 | 0.12 | 0.10 | 0.04 | -0.39 | -0.47 | 0.15 | | |
| *NAR* | 0.21 | 0.21 | -0.85 | -0.86 | -0.83 | -0.03 | -0.09 | -0.63 | 0.08 | |


Because of significant temporal trends in the size of "reticulofenestrids" (Guitián et al., 2020), temporal variations in the thickness of coccoliths may reflect changes in population mean size. The temporal variations in thickness are uncorrelated with the shape factors $k_s$ or $k_{se}$ (r <|0.05| and p>0.1), suggesting different controls on the two indices. Trends in thickness of the whole size population are dominantly an expression of changing coccolith length; the temporal changes in thickness are

highly correlated (r=0.85) with variation in the coccolith length. Even when considering only the population of coccoliths with lengths between 3.5 and 4.5 μm, the temporal trend in thickness still shows significant modest correlation (r=0.63) with the length. For this subset the common effect of shape on thickness is evident: the thickness of coccoliths is modestly correlated with the $k_{se}$, and strongly correlated (r=0.81) with the $k_{se}$ of the full population of coccoliths.

Several factors may contribute to temporal changes in $k_{se}$: (a) the primary coccolith biomineralization, (b) secondary

dissolution that might reduce the coccolith thickness or preferentially dissolve small coccoliths, and (c) secondary overgrowth which might increase coccolith thickness. Because the carbonate content is low at IODP Site 1406 due to the high detrital clay and silt components, preservation of nannofossils is generally excellent (Norris et al., 2014), with no significant overgrowth described. In pelagic settings, low $CaCO_3$ % can be an indicator of dissolution, and a positive correlation between the $k_{se}$ and $CaCO_3$ content of sediment might be expected if dissolution were a significant control on

both parameters, or if higher $CaCO_3$ content in sediment conditions greater diagenetic overgrowth. We find no significant correlation between the $k_{se}$ and $CaCO_3$ content (Table 2). However, the $CaCO_3$ ranges only between 35% and 46% in our examined samples, and these minor variations may be controlled not by dissolution but rather by dilution with detrital minerals, from either lateral transport or rivers draining to the shelf. Nannofossil Accumulation Rate (NAR) is inversely correlated with thickness but not $k_{se}$. The inverse correlation of NAR and length suggest that NAR is driven by the coccolith

size (lower accumulation rate when large cells dominate the population). We suggest that for IODP Site 1406 sediments, the nannofossil assemblage may be the most robust dissolution indicator, because nannoliths such as *Discoaster* and *Sphenolithus* are more dissolution resistant than placoliths like *Reticulofenestra* and *Coccolithus* (Gibbs et al., 2004). The following nannoliths/placolith ratio was calculated:

$$\frac{\% \, Spenolithus + \% \, Discoaster}{\% \, Reticulofenestra + \% \, Coccolithus} \qquad (3)$$

This ratio is strongly inversely correlated with the $k_{se}$ (Table 2; r= -0.64) indicating high relative abundance of nannoliths coincides with reduced calcification volume, potentially consistent with increased dissolution thinning coccoliths. The highest nannolith/placolith ratio, and most intense dissolution, is inferred from the Early Miocene (here 18.4 to 18.8 Ma on





the timescale of (Guitián et al., 2019). This interval has been recognized as a significant hiatus in carbonate rich pelagic sections globally, inferred to reflect a global dissolution event (Sibert and Rubin, 2021). We therefore cautiously consider

most robust the temporal variations in coccolith thickness which occur in samples with low relative abundance of nannoliths (e.g. <2%; for which the correlation between nannoliths percent and $k_{se}$ becomes insignificant).

### 4.4 Temporal trends in *Reticulofenestra* calcification and relationship to environmental factors

The $k_{se}$ decreases from Early to Mid-Oligocene, then progressively increases to attain 30% greater $k_{se}$ in the Early Miocene 18 Ma sample (Figure 6). The Oligocene-Miocene Transition (23 Ma) exhibits a local maximum in $k_{se}$. The Middle

Oligocene minimum $k_{se}$ coincide with the lowest $U_{37}^{k'}$ Sea Surface Temperature (SST) (Guitián et al., 2019), but overall there is not a strong correlation between $k_{se}$ and SST; for example similar low SST in the Early Miocene around 20 Ma coincide with significantly higher $k_{se}$. These changes in $k_{se}$ reflect changes in the $CaCO_3$ content of coccoliths, which are not dependent on size scaling of the lith or cell.

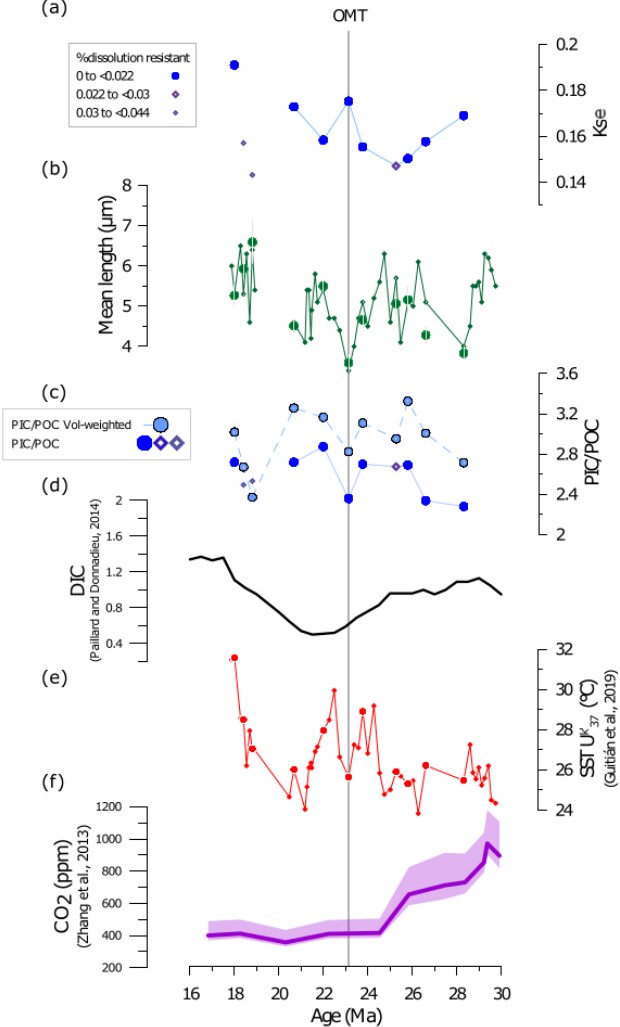





**Figure 6**. Temporal trend in *Reticulofenestra* median coccolith $k_{se}$ vs other morphological and environmental parameters. (**a**) Median *Reticulofenestra* $k_{se}$. Symbols are coded to indicate the relative abundance of dissolution resistant nannofossils, with largest symbols denoting the populations which may be least affected by dissolution; (**b**) length of samples examined in this study (median, large symbol) and mean size in a higher resolution previously published (Guitián et al., 2019, 2020) in small

symbols; (**c**) estimated PIC/POC ratio for the assumptions described in the text. The darker symbol indicates the calculation for the median thickness and length, with symbol size coded to indicate the relative abundance of dissolution resistant nannofossils. The lighter symbol color denotes the estimated population PIC/POC which is weighted by the size distribution as discussed in Guitián et al., (2020); (**d**) estimated ocean DIC concentration from Paillard and Donnadieu, (2014); (**e**) Sea surface temperature (SST) estimated from alkenone unsaturation index, from Guitián et al., (2019); (**f**) estimated atmospheric $CO_2$ from Zhang et al., (2013).

The factors driving changes in $k_{se}$ remain under investigation from a variety of approaches that bear on different aspects of calcification, considering both the organism level factors promoting calcification to the ecosystem scale costs and benefits of calcification. At the organism level, from an experimental manipulation approach, culture experiments examine the response of clonal strains to a range of potential variables, which may be individually manipulated. Meta-analysis of a wide array of *E. huxleyi* culture experiments suggests that high pH and high $HCO_3^-$ concentration both stimulate higher calcification rates

(Bach et al., 2015); calcification rate is dependent on the cell division rate as well as the degree of cellular calcification. Consistent with this evidence, the degree of calcification, expressed as calcite per cell surface area, at constant $CO_{2[aq]}$, increased with both increasing pH and increased alkalinity in culture experiments of *E. huxleyi* (Bach et al. (2015); see Figure S9 in Bolton et al. (2016)). One potential caveat is that the morphology of different coccolith strains differs, and if the morphology of coccoliths is genetically controlled, such experiments may exhibit more limited plasticity compared to that

observed in diverse populations in which selection also influences calcification. From isotopic tracers and observation of long-term decline in cellular calcification and $CO_2$ over the last 14 million years, it has been proposed that when $CO_{2[aq]}$ is limiting, calcification may be reduced because there may be a competitive reallocation of $HCO_3^-$ from calcification to carbon concentrating mechanisms supporting photosynthesis (Bolton et al., 2016). However, the potential role for changing dissolved inorganic carbon (DIC) or $HCO_3^-$ concentrations was not independently constrained in this study. Collectively,

these studies generally suggest that higher $k_{se}$ values may be favored by higher seawater pH, and/or higher DIC or $HCO_3^-$, and/or higher $CO_{2[aq]}$. Alkenone-based $pCO_2$ records have suggested a $pCO_2$ decline from the Early Oligocene to the Early Miocene (Zhang et al., 2013), so the general increase in $k_{se}$ is not consistent with it according to published $pCO_2$ proxies (Figure 6). Estimates of seawater pH from boron isotopes do not span this Oligocene interval (Rae et al., 2021) so cannot be used to test if an increase in seawater pH coincides with increased $k_{se}$. Estimations of DIC from the amplitude of deep ocean

carbon isotopes are low resolution and suggest minimum DIC in the latest Oligocene and an increase into the Early Miocene, only partially matching the trends in thickness data (Paillard and Donnadieu, 2014).

    At the ecosystem scale, the advantages and costs of calcification may also exert selective pressure on the $k_{se}$. High calcification is a potential cost for large coccolithophores sinking out of the euphotic zone in weakly mixed environments (Monteiro et al., 2016). The large (7-9 µm major axis) Oligocene *Reticulofenestra* have $k_s$ in the range of 0.06, similar to

modern $k_s$ (0.06 to 0.08) calculated for large *Coccolithus pelagicus* or *Calcidiscus leptoporus* (Young and Ziveri, 2000), suggestive of similar sinking velocities.

    Finally, we examine the temporal variation in $k_{se}$ in the context of inferred variation in lith and cell size (Figure 7). A null hypothesis is that size does not affect shape; i.e. the coccolith of a given shape has an identical $k_s$ or $k_{se}$ regardless of length. However, adaptation of cells to the biological advantages and disadvantages of larger or smaller sizes and concomitant

changes in surface area to volume ratios may entail adaptation of the degree of calcification, which may be manifest as a trend in $k_{se}$ with size. Comparing the median $k_{se}$ and length of each of the 11 samples, we observe a general decrease in $k_{se}$





with an increase in mean length (Figure 7c); with the 18 Ma as clear exception. This relationship suggests that populations with larger cells have decreased the degree of calcification. Previously, coccolith length in this site was hypothesized to respond directly to the $CO_2$ selective pressure, with larger coccoliths and cell sizes facilitated by higher $CO_2$ concentrations (Guitián et al., 2020). If this interpretation is correct, it might imply decreased calcification under periods of higher $CO_2$, but a more rigorous evaluation would require a direct $pCO_2$ indicator.

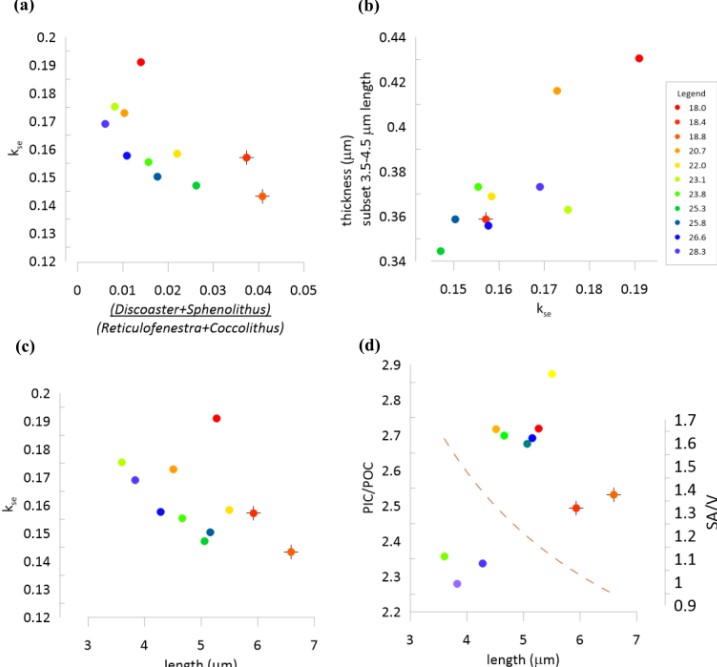

**Figure 7.** Relationship among median "reticulofenestrids" morphological parameters of the 11 populations examined. (**a**) Relative abundance of dissolution resistant taxa vs. *Reticulofenestra* $k_{se}$; (**b**) median thickness of all *Reticulofenestra* with length between 3.5 and 4.5 microns, and the $k_{se}$; (**c**) median $k_{se}$ vs median length; (**d**) estimated PIC/POC ratio for the assumptions described in the text, vs length. The dashed brown line indicates the decrease in surface area to volume (SA/V) ratio of a spherical cell corresponding to the diameter estimated from the median lith length. In all panels, the symbol color indicates the age of examined sediment. The black cross beneath the symbol indicate samples with dissolution resistant taxa exceeding 3%.

**4.5 Implication of variable PIC/POC ratio of dominant coccolithophores**

We evaluate whether data on the coccolith thickness and shape factor can be used to estimate temporal trends in the ratio of particulate inorganic (calcite) carbon to organic carbon (PIC/POC) of coccolithophores. The PIC/POC ratio is surmised to be important factor determining the role of coccolith export fluxes in the ocean's carbon cycle (Barker et al., 2003). Additionally, it has been proposed to be an important factor in the interpretation of coccolithophore-derived geochemical proxies (Mcclelland et al., 2017). It is difficult to estimate the PIC/POC from fossil assemblages in which coccoliths are almost universally detached from the coccosphere, but we examine possible constraints.



Because coccolithophores are not characterized by large vacuoles, the organic carbon quota can be estimated from the cell volume, in turn estimated from coccolith length. Here, we apply the formula to estimate cell diameter from coccolith length (Henderiks, 2008):

$$Diameter = 0.55 + 0.88 * length \qquad (4)$$

We employ the regression for organic carbon from:

$$POC = 0.284 * volume^{0.875} \qquad (5)$$

Generally, as cell size increases in a given genera, it is the size of the coccoliths, rather than the number of liths per cell, which increases (Henderiks, 2008). Consequently, if the number of coccoliths per cell were relatively constant on fossil

*Reticulofenestra*, then the PIC per cell would be the product of the mass of each individual coccolith and the number of coccoliths per cell. This is consistent with the correlation of coccolith thickness and PIC/cell in cultures of extant coccolithophores (Bolton et al., 2016). Our absolute values of PIC per cell are derived from assuming 12 coccoliths per cell, within the range of extant *Gephyrocapsa*, but use of a different number would not change the trends. In this way, we estimate the trend of variation in median PIC/POC ratios resulting from the variation in inferred cell size and coccolith mass

(Figure 6). Additionally, to give account for the disproportionate impact of larger cells on the population, we also calculate a cell volume-weighted median PIC/POC as was previously done for coccolith mean length (Guitián et al., 2020).

According to this estimation, the variations in *Reticulofenestra* PIC/POC were modest, ranging from 2.28 to 2.87, indicating 26% higher PIC/POC in the Early Miocene than in Middle Oligocene. Lowest PIC/POC is inferred for the cold periods of the Middle Oligocene Glacial Interval and the OMT (Figures 6 & 7). Among the populations featuring limited dissolution

(% nannolith/placolith ≤2.2%), the mean estimated PIC/POC correlates positively with major axis length. If covered by a uniform thickness of calcite, larger cells would have a lower (PIC/POC), owing to the 30% decrease in cell surface area to volume ratios over this cell diameter range, a response observed in some clonal culture experiments in which coccolith morphology varies little (Bolton et al., 2016). These results highlights that coccolith thickness estimates, in addition to cell diameter, is required to predict PIC/POC ratios of fossil communities.

**5. Conclusions**

We implement a precise and reproducible calibration routine in the C-Calcita software to enable robust estimations of coccolith thickness over the range 0 to 3.1 microns, the largest calibration range of any technique yet reported. This approach can be applied to any microscope configuration with circular polarization when a suitable range of calibration points is defined on a calcite wedge.

Application of this method provides the first evaluation of the thickness and shape factor of coccoliths from the Oligocene to Early Miocene. In this age interval, we find that median thickness is highly correlated with coccolith length, and we adopt the elliptical shape factor ($k_{se}$) as a metric which more accurately describes the differences in the degree of calcification of a coccolith of given length. The $k_{se}$ is inversely correlated with the relative abundance of dissolution resistant nannoliths, suggesting that greater dissolution may lead to partial dissolution of coccoliths and lower shape factors. Therefore, we

suggest that future evaluations of coccolith thickness and $k_{se}$ should include assessment of variation in dissolution intensity. By comparing $k_{se}$ among samples which likely experienced similar dissolution intensity, this study highlights for the first time a significant evolution in the degree of calcification of coccolithophores of given sizes. We identify the Middle Oligocene was characterized by the lowest $k_{se}$ and the late Early Miocene was characterized by the highest $k_{se}$. While the Miocene peak is coincident with the maximum in inferred ocean DIC concentration, the long term trend appear to correlate

with size decrease, and not in agreement with the existing DIC or $CO_{2[aq]}$ variation, thus a relationship worth further exploration.



**Data availability**

Data generated for this study is presented in Table 1 and will be uploaded to Zenodo public repository during the review process (https://doi.org/10.5281/zenodo.6341696).

**Acknowledgments**

Sediment samples were provided by the Ocean Drilling Program (ODP). Authors thank Remy Lüchinger from ETH Zurich for calcite wedge manufacturing. This study was supported by the Swiss National Science Foundation (Award 200021_182070 to Heather Stoll).

**Author contributions**

MA developed the algorithms, implemented them in C-Calcita and applied them to classified coccoliths. JG prepared and calibrated the calcite wedge, prepared slides, took images and classified coccoliths for processing. IH oversaw processing of some additional samples, assisted with statistics and elaboration of figures. HS conceived of the calcite wedge calibration, interpreted the morphometric data resulting from the application of the algorithm. HS and JG wrote the paper with input of text from MA and feedback from all authors.

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
