# Peer review of "Variation in calcification of *Reticulofenestra* coccoliths over the Oligocene-Early Miocene"

_Biogeosciences, 2022_

## Referee Comment (RC1)

I have read with great interest the paper of Guitián et al. on the "Variation in calcification of *Reticulofenestra* coccolith over the Oligocene-Early Miocene". The article presents new data, and the authors might be right in stating that this is one of the first papers of its kind in the studied region. The article merits publication in *Biogeosciences*. And I would suggest some minor corrections.

Specifically, I have the following comments:

General about *Reticulofenestrids*,

In the Early Miocene, the assemblages of Noelaerhabdaceae include many *Cyclicargolithus floridanus*, which are circular shapes. Therefore, I suggest that this specimen (or group) is separated from *Reticulofenestra* group.

Chapter 2

Line 88; How do you estimate the thickness and volume between proximal and distal shields? Please add the details of the analysis method of thickness under the microscope.

Chapter 4

Line 180; It seems that these thicknesses indicate only distal shield size. The reticulofenestrids are placolith groups. How do you analyze the thickness between distal and proximal shields?

Table 1; Please adjust the width size of each cells (e.g. centralizing etc.).

Table 2; Please adjust the width size of each cells (e.g. centralizing etc.).

---

## Author Comment (AC1)

We would like to thank the Anonymous Referee #2 for the valuable comments as well the editor for the time considering this manuscript. Below we provide a response to the comments.

Anonymous Referee #2

I have read with great interest the paper of Guitián et al. on the "Variation in calcification of *Reticulofenestra* coccolith over the Oligocene-Early Miocene". The article presents new data, and the authors might be right in stating that this is one of the first papers of its kind in the studied region. The article merits publication in *Biogeosciences*. And I would suggest some minor corrections.
Specifically, I have the following comments:

General about *Reticulofenestrids*,

In the Early Miocene, the assemblages of Noelaerhabdaceae include many *Cyclicargolithus floridanus*, which are circular shapes. Therefore, I suggest that this specimen (or group) is separated from *Reticulofenestra* group.

The reviewer suggests that the assemblage composition of the Noelaerhabdaceae should differentiate the *C. floridanus group* from the *Reticulofenestra* group given its different morphology. However, the difference in shape (circular or elliptical) does not affect the calculation of calcification or shape factor in this study, because unlike previous studies which use a shape factor only based on length (e.g Young and Ziveri, 2000), here we use both the length and width, effectively accounting for changes in the circularity. We consider that for an appropriate environmental survey of the Noelaerhabdaceae family and the scope of this study (evaluate changes in cellular calcification of main coccolithophores from the Oligocene-Miocene time interval); all groups from the *Reticulofenestra* genus (i.e *R.bisecta*, *R. lockeri*, *R. umbilicus*, *C. floridanuds, etc*) should be taken into account. Importantly, the high calcifying *C. floridanus* group should be included given its important contribution to the carbonate fraction. This is now clarified within lines 52 to 54 and through the text.

Chapter 2

Line 88; How do you estimate the thickness and volume between proximal and distal shields? Please add the details of the analysis method of thickness under the microscope.

Chapter 4

Line 180; It seems that these thicknesses indicate only distal shield size. The reticulofenestrids are placolith groups. How do you analyze the thickness between distal and proximal shields?

We thank the reviewer for prompting us to further clarify components quantified in the thickness calculation. The measured thickness corresponds indeed to the total accumulated calcite thought the crystal (i.e. proximal shield + distal shield). Using circular polarization, all the birefringent materials (r-units) (Young et al., 2004) are seen and present no extinction at any orientation. Thus, the components of the coccoliths formed by r-units (shields, tubing, etc.) are seen with a light intensity and colour that are determined by the thickness of the calcite. Now we clarify this in the text lines 87-90 and to avoid misunderstanding also at figures 2 and 3 footnotes.

Table 1; Please adjust the width size of each cells (e.g. centralizing etc.).

Table has been adjusted and will be presented in the supplement as table S1 following referee 2 comments.

Table 2; Please adjust the width size of each cells (e.g. centralizing etc.).

Table has been modified into new figure 6 where we show all correlation coefficients in a clarified format

References

Young, J. R. and Ziveri, P.: Calculation of coccolith volume and it use in calibration of carbonate flux estimates, Deep sea research Part II: Topical studies in oceanography, 47, 1679-1700, 2000

Young, J. R., Henriksen, K. & Probert, I. (2004). Structure and morphogenesis of the coccoliths of the CODENET species. In, Thierstein, H. R. & Young, J. R. (eds) Coccolithophores - From molecular processes to global impact. Springer, Berlin 191-216.

---

## Author Comment (AC2)

**Comment on bg-2022-66**
Anonymous Referee #3
Referee comment on "Variation in calcification of *Reticulofenestra* coccoliths over the Oligocene-Early Miocene" by José Guitián et al., Biogeosciences Discuss., https://doi.org/10.5194/bg-2022-66-RC2, 2022

I enjoyed reviewing the paper by Guitián et al., I'm looking forward to seeing the paper published and I suggest a moderate/minor revision before being suitable for publication. The paper is mostly well written however here and there some sentences are not hanging and as if the authors don't feel like getting deep into the discussions.
I also suggest reviewing some of the Figures that are difficult to follow.
Down there are my comments:
We would like to thank the comments carefully provided by Anonymous Referee #3 that substantially improved the manuscript. We have addressed the issues noted by clarifying statements and improving graphs and tables. Also the results description and evaluations described in section 4.2 about the relationship between size and calcification have been improved, as well the discussion of trendlines in Section 4.

Below we provide a response to comments in red.

**Abstract**
Line 12: the coccoliths and,
Text adjusted

Line 14: the ocean carbon system is not an environmental parameter. I understand what the authors are referring to, but this sentence should be rephrased.
Rephrased. Now states, "An open question is whether their calcification has been affected by changing environmental conditions over geological timescales such as variations in the ocean carbon system"

**Introduction**

Line 34: "cellular growth and degree of cellular calcification" both need some citations (that is not Bolton et al., 2016 which is already cited many times in the text). Line 35: do you mean that the cellular calcification is controlled by the thickness of coccoliths? The way it is written seems to me that also the growth rate is controlled by coccolith thickness. Add: the latter is first order controlled…".
We have adjusted the sentence for clarity. A rate of calcification is necessarily the product of the rate of cell division and the amount of calcite per cell and this fact is generally not provided with citations in the culture literature. We now have separated the sentences and moves the second sentence to later in the paragraph. First we state "Calcification rates of coccolithophores depend on both the cellular growth rate and the degree of cellular calcification". Then by new line 43 we state: "Culture studies show that the degree of cellular calcification is to first order controlled by the thickness of coccoliths (Bolton et al., 2016)". Bolton et al., (2016) is indeed the correct citation, as it presents the relationship between the thickness of coccoliths and cellular calcification across a number of strains of extant Noelaerhabdaceae.

Line 38: $CO_2$ is one of the media carbonate chemistry parameters. I would delete $CO_2$ and leave "changing carbonate chemistry".
Done

Line 39-44 I think that the aim of the paper should be better expressed here. It is only partially clear that you would like to test whether variation in cellular calcification due to $CO_2$ variations is transient or last longer in geological times. I guess that the final aim is

to be able with these data/analyses to reconstruct past ocean CO2? It would be important that the main scientific questions are better expressed in this paragraph.

Paragraph has been rewritten highlighting the main questions for this study.

Line 55: check on Copernicus's rules on how to cite these papers. Maybe: (1.55 mm, Bollmann, 2014, Fuertes et al., 2014; …). Not sure but please check if the brackets are correct. And more in general check in the whole text.

Corrected, brackets added outside.

**Approaches for estimation of coccolith thickness**
Line 68: cellular calcification: do you mean the thickness?

Sentence has been clarified. Cited study shows cellular calcification decrease, based on the coccolith thickness.

Line 72 delete:

Done

Line 80: following Young (1998)

Deleted in the revised version.

Line 89: (e.g. Fuertes et al., 2014)

Done

**Calibration of absolute…**
Line 103: check if it is correct to write the reference in this way. Should it be: described in Golnzalez Lemos et al. (2018)?

Done

Line 104: Figure 1 in brackets.

Done

Line 107: "as similar field of view as possible" or "as a similar field of view as possible"

Clarified. "as similar field of view as possible".

**3.3 Calculation of**
Line 136: check if it is correct to write the reference in this way. Should it be by Flores and Sierro (1997)?

Done

**Indices of variations…**
Line 158: delete double brackets

Done

**4.1 Thickness of Oligocene nannofossils**
Figure 4a: in the caption or in the figure, the authors should mention what the different names referred to. E.g. retics = Reticulofenestra; pelag = Coccolithus and so on.

Done, including *Helicosphaera* which is also plotted.

Line 199-202 and Figure 4 + caption: I cannot find Figure 4c and I think the authors are referring to Figure 4a. I suggest renaming every graph as 4a,b,c,d so that it is easier to follow.
As I mentioned in the previous comment would be also important to make explicit the abbreviations (e.g. pelag). Moreover in Fig. 4b if the values refer to a species would be ideal to add the full name for Coccolithus and Discoaster (e.g. Coccolithus pelagicus). Use L instead of major axis (as explained in the text) or be consistent in the text and in the graph using Length or L or major axis.

Figure is now renamed as suggested by reviewer and figure is cited in the text accordingly. Text has been revised and term length is used instead of major axis for the maximum dimension of the coccolith plate (as stated in line 168: "we refer to the major axis as length (l) and the minor axis as width (w)".

Line 210: "future studies should be interpreted": I don't get it: instead of what. Size?
We now clarify that for those cases of non-retics data, results should be interpreted as relative change in thickness instead of change in total accumulated coccolith thickness (and therein, calcification)

**4.2 Relationship between…**
Line 215: it would be nice to add a table with all values e.g. mean, median sd deviation
more complete than table 10, maybe in the supplementary.
We have expanded previous Table 1, adding referee suggestions and is now presented as supplement table S1.

Line 216: is this increase in ks correlated with variations in size significant? Could you add the r2 values in the plots (since you represented already the trend lines)?
Figure 5 and caption: You should add in the caption or in the figure the number of specimens and of samples measured for each time interval. You should choose also here if you want to use "major axis" or l or Length.
In the new version, figure 5 is updated showing r2 values and number of specimens measured

Line 218: Do you mean that there are no data available on Ks on large populations of extant Noelaerhabdaceae? If yes, could you rephrase and avoid to you: "we are aware".
In general, I don't get the main message in this chapter: the main issue is that the authors found mainly stable Ks. But when Ks is bigger, the length of *Reticulofenestra* is smaller. First of all, would be important in Figure 5 to have also one graph with all values together and not separated into different intervals. Would be also nice to have a regression line and r2.
The second key message of this chapter is that studies on living coccolithophores don't show the same relation but indeed the opposite one (> L and > Ks). I think this part could be implemented because the chapter stops with two pending sentences that don't explain this opposite trend. Especially the last sentence: I don't understand why it is written here at the end of the paragraph and what the author wanted to explain/evidence. All population refers to what? Do you mean in the different intervals? Are you referring only to *Reticulofenestra*? I guess yes due to the name of the chapter and also looking at Fig. 4b (thickness versus major axis). Can you add the slopes and calculate r2 for every group in Figure 4b?
The paragraph has been revised and reorganized in order to highlight our findings, where the relationship between length and calcification-related variables (thickness, shape factors).
We distinguish that thickness is positively correlated with length, while the scale invariant shape factor Kse and length are not strongly correlated. We have expanded and clarified the comparison with data on extant taxa,
    a)  As before, we comment the contrasting result with cultures in which ks were determined on large populations of *Reticulofenestra* coccoliths from different strains.
    b)  We expand the information provided on the study comparing different modern (*Reticulofenestra* and non-*Reticulofenestra*) placolith taxa (Young and Ziveri, 2000), highlighting the range of generally positive covariation in ks and length and the range of inverse correlation of length and ks at larger lengths. This detail replaces our previous observation that positive correlation between ks and length was "not universal" citing the Young and Ziveri (2000) study.

Following referee suggestion, we have also added to Figure 5 two graphs with the relationship between ks, kse and length for all the samples pooled together, with their corresponding trend lines. Additionally, in Figure 4 (now Figure 4e) we have added the regression statistic and fit lines for each of the three populations of coccoliths plotted.

**4.3 Indicators of coccolith calcification and relationship to the preservation**
Line 243: r=0.63 to what it refers to? Is this value represented in Table 2? Could you otherwise add it since it refers only to small coccoliths? I think that in general, it's very hard to follow Table 2 so I might misunderstand this part. The abbreviations need to be explicit. Could you make them explicit in the caption?
In order to make previous Table 2 correlation values more understandable, we have modified it to a new figure with color shading for the strength of correlations (Figure 6). Labels have been modified. For the small coccoliths subset, we now added a table with this data in the excel supplement as Table S2.

Line 251: this paragraph has been copied twice (see line 237) and needs to be deleted.
Paragraph duplicate has been deleted

Table 10: caption: "Samples, ages and the values of the median values of the measured morphological parameters". Could you also add for each sample the number of measured specimens?
Table has been updated and presented now as Table S1 including all referee suggestions.

Line 269: NAR is also inversely correlated with the length suggesting a lower accumulation rate when large cells…
Sentence has been corrected

Line 275: I would be more cautious and delete strongly for -0.64.
We have updated the sentence which now states. "This ratio is inversely correlated with the kse (Figure 6) indicating…"

Line 278: check if the reference is written in the right way following Copernicus regulations.
Reference now in the correct format

**4.4 Temporal trends in *Reticulofenestra* calcification and relationship to environmental factors**
Figure 6: can you add the series in the figure?
The symbols used in this Figure are hard to follow. The authors should use different colours instead of different types of symbols (or in combination different colours and symbols). For sure they have to avoid using the same symbol and colour with smaller sizes of the symbols. This refers to a and c.
For figure b, why is there a hiatus in size trend in the Miocene?
For Figure a, the line should be deleted or used for all the sets of data. Otherwise, this representation is misleading.
In the new version of the manuscript we provide an updated version of the figure (new Figure 7) following referee suggestions. Here, we believe color and size symbol code used are more appropriate. The lack of line in the Miocene was acknowledging a hiatus present in the core sediment described in previous studies (Van Peer et al., 2017), however, we understand that this might disorient the reading of the results and thereby we updated the figure with trendlines connected for all graphs and appropriate description at the caption. We hope these revisions attain the improvements proposed by the reviewer.

Figure e and f: are these two graphs correlated in the right way? CO2 and temperature seem to go in opposite directions which surprises me a lot. Maybe it's a matter of resolution, but these two graphs one next to the other seem to say two different stories.
The graphs accurately plot published CO2 proxy records and temperature records; the relationship among these proxies in the Oligocene and early Miocene is under investigation (see also discussion in Guitian et al., 2020)

Line 287: when you use the word significantly does it mean that there's a statistical

analysis behind it? Indeed, it would be important, if you have the data of Ks and the other parameters for the same sample, to statistically correlate them.

The relationship between the sea surface temperature from Guitian et al., 2019 and the Kse/ks factors at the same samples is now presented in the text as the r value = 0.46 and terms revised accordingly (new line 281). The rest of parameters for each sample are statistically related at new Figure 6.

Lines 283-285: this description is not really accurate. It's true only the authors avoid to considers the two samples with lower dissolution resistant species. Moreover, the point just after the OMT exhibit similar values compared to the Early to Middle-Oligocene sample. Should be carefully re-written.

We have reorganized the statements on this paragraph to better describe the temporal evolution of the shape factors presented here.

Line 315: these studies: which one?

We have clarified that we refer to the culture and Miocene to present observations

Line 324-326: This sentence is pending. I mean: what do the authors mean with this paragraph?

Paragraph has been re-written for clarification improving reasoning about meaning of potential similar sinking velocities.

**References**

Bolton, C. T., Hernández-Sánchez, M. T., Fuertes, M. Á., González-Lemos, S., Abrevaya, L., Mendez-Vicente, A., ... & Stoll, H. M. (2016). Decrease in coccolithophore calcification and CO2 since the middle Miocene. Nature communications, 7(1), 1-13.

Guitián, J., Dunkley Jones, T., Hernández-Almeida, I., Löffel, T., & Stoll, H. M. (2020). Adaptations of coccolithophore size to selective pressures during the Oligocene to Early Miocene high CO2 world. *Paleoceanography and Paleoclimatology*, *35*(12), e2020PA003918.

Van Peer, T. E., Xuan, C., Lippert, P. C., Liebrand, D., Agnini, C., & Wilson, P. A. (2017). Extracting a detailed magnetostratigraphy from weakly magnetized, Oligocene to early Miocene sediment drifts recovered at IODP Site U1406 (Newfoundland margin, northwest Atlantic Ocean). Geochemistry, Geophysics, Geosystems, 18, 3910–3928. https://doi.org/10.1002/2017GC007185

Young, J. R. and Ziveri, P.: Calculation of coccolith volume and it use in calibration of carbonate flux estimates, Deep sea research Part II: Topical studies in oceanography, 47, 1679-1700, 2000

---

## Author Response (AR2)

**Comments to the author**:

Dear Dr. Guitián and Co-Workers,

it is my pleasure to accept your revised manuscript subject to technical corrections. These corrections are mainly associated with the incompatibility of some of your figures for persons with red-green color blindness (please read also our guidelines for figure preparations).

Fig. 1 (d) 1: I understand this is a color band spectrum and it is difficult to change the colors. But please make sure the red and the green graph are distinguishable (maybe label the graphs).

Fig. 4 b and e: The green- and orange-filled circles are indistinguishable. Please use either different symbols or compatible colors.

Fig 7: The revised version of the graphs seems to be missing the letters (a-f)

Fig. 8: Please do not combine red/orange/green. You can keep the colors if you assign different shapes of symbols.

Data availability: The doi link to the data does not work. I assume it has not been released yet. Please make sure the doi is active upon publication of the manuscript.

Best
Tina Treude

Dear Editor,

In the new version of the manuscript we have corrected the color incompatibilities of figures 1, 4 and 8 and added the missing letters to figure 7. Regarding the data availability, the doi link to zenodo archive is now active (10.5281/zenodo.6341696).

Best,

Jose Guitian